# Hallucinating brains with artificial brains

## Abstract

Human brain function as measured by functional magnetic resonance imaging (fMRI), exhibits a rich diversity. In response, understanding the individual variability of brain function and its association with behavior has become one of the major concerns in modern cognitive neuroscience. Our work is motivated by the view that generative models provide a useful tool for understanding this variability. To this end, this manuscript presents two novel generative models trained on real neuroimaging data which synthesize task-dependent functional brain images. Brain images are high dimensional tensors which exhibit structured spatial correlations. Thus, both models are 3D conditional Generative Adversarial networks (GANs) which apply Convolutional Neural Networks (CNNs) to learn an abstraction of brain image representations. Our results show that the generated brain images are diverse, yet task dependent. In addition to qualitative evaluation, we utilize the generated synthetic brain volumes as additional training data to improve downstream fMRI classifiers (also known as decoding, or brain reading). Our approach achieves significant improvements for a variety of datasets, classification tasks and evaluation scores. Our classification results provide a quantitative evaluation of the quality of the generated images, and also serve as an additional contribution of this manuscript.

## 1 Introduction

Functional Magnetic Resonance Imaging (fMRI) is a common tool used by cognitive neuroscientists to investigate the properties of brain function in response to stimuli. Classic analysis approaches (Poldrack et al., 2011) focused on analyzing group-averaged brain function images. However, it was discovered that brain activation patterns vary significantly between individuals. Thus, modern analysis now prioritizes understanding the inter-subject variability of brain function (Dubois & Adolphs, 2016; Geerligs et al., 2017). Our work is motivated by the view that generative models provide a useful tool for understanding this variability – as they enable the synthesis of a variety of plausible brain images representing different hypothesized individuals, and high-quality generative models can be analyzed to posit potential mechanisms that explain this variability (Horn et al., 2008). The results presented in this paper provide – to our knowledge for the first time, positive results suggesting that it is indeed possible to generate high quality diverse and task dependent brain images.

While we can qualitatively evaluate generative brain images, quantitative evaluation allows us to objectively compare between various results. To this end, we utilize the generated synthetic brain volumes as additional training data to improve downstream fMRI classifiers. The use of classifiers to predict behavior associated with brain images is also known as decoding or brain reading (Varoquaux & Thirion, 2014; Pereira et al., 2009). Classifiers such as support vector machines and deep networks have been applied for decoding brain images. For example, Cox & Savoy (2003) attempted to classify which of 10 categories of object a subject was looking at (including similar categories, such as horses and cows) based on limited number of brain images. Besides visual tasks, Mitchell et al. (2008) distinguished active regions of brains when subjects listened to linguistic words, where the stimuli included five items from each of 12 semantic categories (animals, body parts etc.).

Beyond providing a model for individual variability, high quality brain image synthesis addresses pressing data issues in cognitive neuroscience. Progress in the computational neurosciences is stifled by the difficulty of obtaining brain data either because of a limited culture of data sharing, or due to medical privacy regulations (Poldrack & Gorgolewski, 2014). For the computational neuroscientist,

generated images deliver *unlimited* quantities of high quality brain imaging data that can be used to develop state of the art tools before application to real subjects and/or patients (Varoquaux & Thirion, 2014). This approach of using modern generative models to synthesize data, which in turn accelerates scientific study, has already proven useful in many scientific fields such as particle physics and astronomy (Castelvecchi et al., 2017). Our work represent a first application for this approach to neuroscience.

One of the promising generative models are Generative Adversarial Networks (GANs) (Goodfellow et al., 2014), capturing complex distributions using a non-cooperative two-player game formulation: a generator produces synthetic data by transforming samples drawn from a simple distribution; a discriminator focuses on distinguishing synthetic and real data. Despite (or possibly due to) the compelling formulation, GAN training is known to be unstable. To address this difficulty various variants have been proposed. Wasserstein GANs (WGANs) (Arjovsky et al., 2017) formulate the objective using the Wasserstein distance rather than the classical Jenson-Shannon divergence. Improved training of WGANs (Gulrajani et al., 2017) applies an additional gradient penalty, which avoids the critical weight clipping in WGANs which might lead to pathological behavior. Dualing GANs restrict the discriminator and formulate the dual objective (Li et al., 2017). Beyond improving the stability, conditional GANs (Mirza & Osindero, 2014) make it possible to control the data generation process by conditioning the model on additional information. Auxiliary Classifier GANs (AC-GANs) (Odena et al., 2016) unify a GAN and classifier to a single architecture, employing labels to generate ImageNet samples. 3D GANs (Wu et al., 2016) reconstruct 3D objects and Edward & David (2017) propose to use improved WGAN to enhance the stability of 3D GANs.

We make the following contributions in this paper:
1. We develop Improved Conditional Wasserstein GANs (ICW-GAN) and Auxiliary Classifier and Discriminator GANs (ACD-GAN), two types of 3D conditional GANs to synthesize fMRI brain data, both of which we find to be stable to train.
2. We assess the qualitative quality and diversity of generated brain volumes. Our results suggest that the proposed models are able to generate high quality task-dependent and diverse 3D brain images.
3. We evaluate our models on three datasets using a series of image classification tasks with support vector machines and deep network classifiers at two levels of brain image resolution. Results show that augmenting training data using synthetic data generated by our models can greatly improve test classification accuracy of brain volumes.

## 2 RELATED WORK

### 2.1 NEURAL NETS FOR FMRI BRAIN DATA

We are unaware of any published papers using neural networks to generate brain imaging data. However, neural networks have been used by several authors for classifying brain imaging data. Firat et al. (2014) and Koyamada et al. (2015), used 2D deep nets to extract features of fMRI brain images to classify brain states. Nathawani et al. (2016) applied both 2D and 3D neural networks to classify fMRI brain data. Svanera et al. (2017) decoded fMRI data of video stimuli and classify data into visual categories. Similarly, Nathawani et al. (2016) extracted features from 4-D fMRI data and used deep learning methods for discrimination of cognitive processes.

### 2.2 GENERATIVE ADVERSARIAL NETS

To learn a distribution over data $x$, a GAN (Goodfellow et al., 2014) formulates a 2-player non-cooperative game between two deep nets. The *generator* $G$ uses a random noise vector $z$ sampled from a prior distribution $P_z(z)$ as input and produces an image $G(z)$. The generator is trained to fool the discriminator, which receives either synthetic data or real data and differentiates between them. Formally, $G$ and $D$ play the following two-player minimax game with value function $V(G, D)$:

$$\min_G \max_D V(G, D) := \mathbb{E}_{x \sim P_{data}(x)}[\log D(x)] + \mathbb{E}_{z \sim P_z(z)}[\log(1 - D(G(z)))]. \tag{1}$$

For the conditional GAN (Mirza & Osindero, 2014), both $G$ and $D$ are conditioned on some extra information $y$, for instance, class labels or other features. The conditioning can be presented by feeding $y$ into both the generator and the discriminator. This work is successful in one-to-many mappings such as image labeling with many tags.

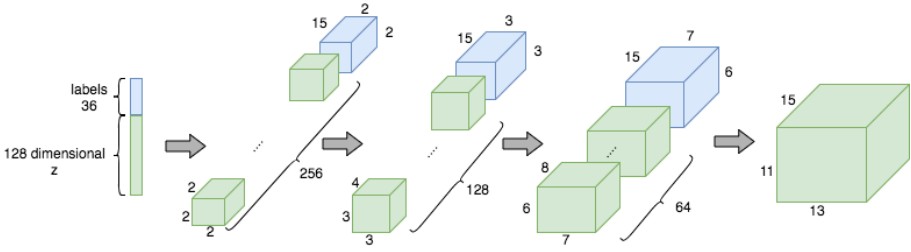

Figure 1: The generator of our ICW-GAN. The 128 dimensional encoding $z$ is drawn from a multi-variate normal distribution. The label vector is a one-hot encoding, i.e., the one entry that equals one represents the class that the generated volume belongs to. It is concatenated to input and hidden layers and for each of these layers, a fully connected layer transforms the label vector to a volume. Our stride in the de-convolutional layers is [1,2,2,2,1] in the batch, height, width, length, feature map dimension.

The Wasserstein GAN (WGAN) (Arjovsky et al., 2017) uses *Wasserstein-1* distance $W(q, p)$ as the objective function. It computes the minimum cost of transporting mass to transform the distribution $q$ into the distribution $p$. Under this circumstance, $W(q, p)$ is continuous everywhere and its gradient with respect to its input was found to be more stable than its classical GAN counterpart. Improved Wasserstein GAN (IWGAN) (Gulrajani et al., 2017) argues that weight clipping of the critic, i.e., the discriminator, in WGANs inevitably causes the gradient to either vanish or to explode. To address this issue Gulrajani et al. (2017) propose an alternative penalty term in the critic loss based on the gradient norm.

In an auxiliary classifier GAN (AC-GAN) (Odena et al., 2016), every generated sample has a corresponding label which can be used as a condition in the *generator* and the *classifier*. The *discriminator* is modified to contain an auxiliary decoder network to reconstruct the training data class. Further, 3D-GANs (Wu et al., 2016) extend GANs to 3D object generation. Different from classical GANs, 3D-GANs apply the three dimensional convolution in both G and the D. By learning deep object representations, 3D GANs can generate visually appealing yet variable 3D object volumes.

## 3 APPROACH

In the following we introduce two models for fMRI data generation: ICW-GAN and ACD-GAN. They differ in model structure and in the way label information is taken into account. We first show the model structure of 3D ICW-GAN and its downstream classifiers in Section 3.1, and present ACD-GAN in Section 3.2.

### 3.1 3D IMPROVED CONDITIONAL WASSERSTEIN GAN (ICW-GAN)

Similar to classical generative adversarial networks (GANs), ICW-GANs are formulated as a non-cooperative two-player game between two adversaries: (1) a generator $\hat{x} = G_\theta(z)$, which generates artificial samples $\hat{x}$ from randomly drawn latent encodings $z$ via a transformation using a deep net parameterized by $\theta$; and (2) a discriminator $D_w(x)$ represented via the logit obtained from a deep net parameterized by $w$.

Following Arjovsky et al. (2017), we build our models based on minimizing the Wasserstein distance via

$$\min_\theta \max_{w \in W} \mathbb{E}_{\boldsymbol{x} \sim P_{data}(\boldsymbol{x})}[D_w(\boldsymbol{x})] - \mathbb{E}_{\boldsymbol{z} \sim P_{\boldsymbol{z}}(\boldsymbol{z})}[D_w((G_\theta(\boldsymbol{z})))], \tag{2}$$

where $\{D_w\}_{w \in W}$ denotes a set of functions that are K-Lipschitz for some K. IW-GANs (Gulrajani et al., 2017) provide a more stable method for enforcing the Lipschitz constraint by employing a gradient penalty rather than weight clipping, when optimizing:

$$\mathbb{E}_{\boldsymbol{z} \sim P_{\boldsymbol{z}}(\boldsymbol{z})}[(1 - D_w(G_\theta(\boldsymbol{z})))] - \mathbb{E}_{\boldsymbol{x} \sim P_{data}(\boldsymbol{x})}[D_w(\boldsymbol{x})] + \lambda \mathbb{E}_{\hat{\boldsymbol{x}} \sim P_{\hat{\boldsymbol{x}}}}[(\| \nabla_{\hat{\boldsymbol{x}}} D_w(\hat{\boldsymbol{x}}) \|_2 - 1)^2]. \tag{3}$$

Hereby $\hat{x}$ is a convex combination of real data and artifical samples, i.e., $\hat{\boldsymbol{x}} \leftarrow \epsilon\boldsymbol{x} + (1 - \epsilon)G(\boldsymbol{z})$ with $\epsilon$ drawn from a uniform distribution ($\epsilon \sim U[0, 1]$).

In our models, we extend IW-GAN in two directions. First, because fMRI data is three dimensional, 3D convolution and deconvolution are used in order to maintain the spatial structure and voxel information of the fMRI data. Second, we condition both the discriminator $D_w$ and the generator $G_\theta$ on available labeled data.

As shown in Figure 1, our generator consists of three fully convolutional layers. In convolutional layers, we use kernels with a size $4 \times 4 \times 4$ and strides 2, with batch normalization and Leaky ReLU

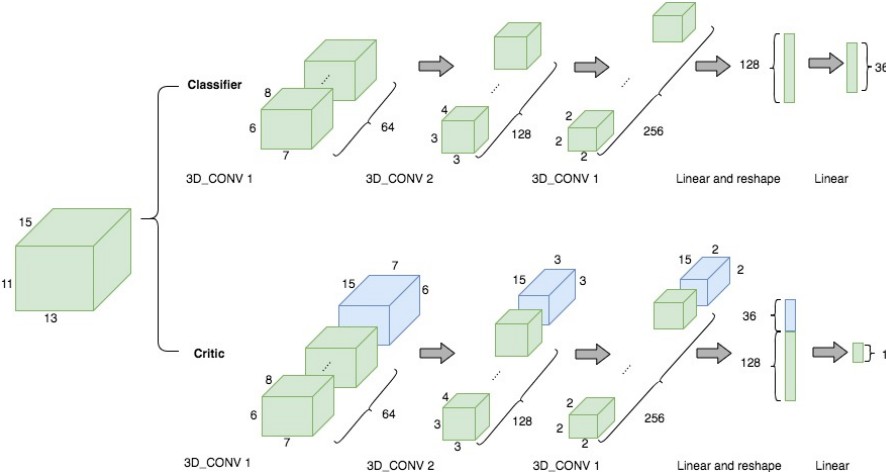

Figure 2: The classifier (top) and critic (bottom) in our ACD-GANs. The classifier and the critic possess a similar architecture but not identical in terms of the output and leveraging of labels. The label is only employed in the critic, not in the classification stream. The way to transform the label vector and the stride are identical to what we illustrated in Figure 1.

layers added between and a tanh layer at the end. The discriminator architecture is a mirrored generator, but no activation function is used at its end.

This ICW-GAN model follows models proposed in (Gulrajani et al., 2017; Mirza & Osindero, 2014; Wu et al., 2016), however, we found the following modifications to stabilize training and supervise the generation process for classification data: compared to existing GANs, our ICW-GAN applies a more stable upper bound for Wasserstein distance as an objective function, leverages conditioning labels to supervise the generation process, and utilizes 3D convolution methods for 3D object generation.

To include label information, we concatenate labels to the input and to the hidden layers. At the input of the generator, one-hot labels are combined with the brain vector. Then, for each of the intermediate layers, we use a fully connected layer followed by a tanh activation to transform the one-hot vector to a volume of appropriate size, i.e., $15 \times 2 \times 2$ for the first hidden layer, and $15 \times 3 \times 3$ for the next. We empirically found dimensions other than 15, e.g., 3 to work well too. We concatenate the label volume to intermediate volumes on feature dimension and pass the joint one to the next deconvolution layer. We follow the same procedure in the architecture of the discriminator.

Referring to the volume labels via $\boldsymbol{y}$, the objective function $L$ of our ICW-GAN model is as follows:

$$L = \mathbb{E}_{\boldsymbol{z} \sim P_{\boldsymbol{z}}(\boldsymbol{z})}[(1 - D(G(\boldsymbol{z}|\boldsymbol{y})))] - \mathbb{E}_{\boldsymbol{x} \sim P_{data}(\boldsymbol{x})}[D(\boldsymbol{x}|\boldsymbol{y})] + \lambda \mathbb{E}_{\hat{x} \sim P_{\hat{x}}}[(\| \nabla_{\hat{\boldsymbol{x}}} D(\hat{\boldsymbol{x}}|\boldsymbol{y}) \|_2 - 1)^2], \quad (4)$$

where $\hat{\boldsymbol{x}} \leftarrow \epsilon \boldsymbol{x} + (1 - \epsilon)G(\boldsymbol{z})$, $\epsilon \sim U[0, 1]$. $\lambda$ is a gradient penalty coefficient. We optimize both the discriminator and generator loss using Adam optimizer (Kingma & Ba, 2014).

### DOWNSTREAM CLASSIFIERS FOR ICW-GAN

Our work is partially motivated by the difficulty of acquiring fMRI data, so we consider the task of enlarging the dataset for downstream tasks such as classification, using synthetically generated samples. To assess the suitability of the developed generation procedure, we employ a variety of downstream classification processes. More specifically, we train a classifier with real ('Real') or real plus generated ('Real+G') data. Common classifiers in neuroscience are Support Vector Machines (SVMs) and 3D deep networks. We think usage of both provides valuable insights as their number of trainable parameters differ significantly, i.e., 14 parameters for SVMs and several hundred thousand for our deep networks. To measure the results obtained from different training datasets we use accuracy, macro F1, precision and recall metrics.

**SVM**: We use a simple linear SVM to classify test data and don't extract any intermediate features. Instead, we use raw brain data, vectorized to a 1-dimensional vector for classification to avoid any form of collusion.

**Deep Net**: The deep net structure is similar to the discriminator, having a 3 dimensional structure and identical number of convolution layers with a Leaky ReLU activation. Obviously the classifier doesn't concatenate intermediate and input data with any label information. We provide the respective information for each dataset in the experimental section.

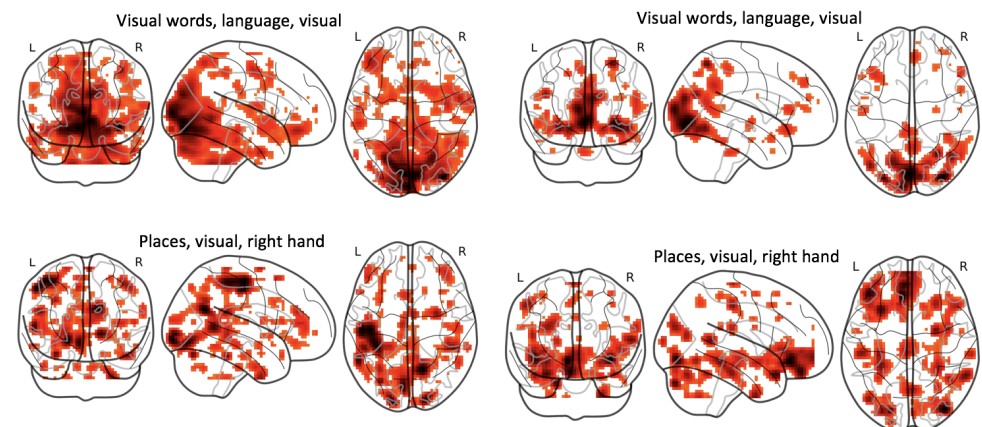

Figure 3: 2D projections of synthetic brain volumes generated using the proposed ICW-GAN. The training dataset is collection 1952. The tag above each image sequence shows the class the picture belongs to. Qualitative evaluation by experts suggests that the generated images are indeed qualitatively realistic.

| Downsampling | Input | Classifier | Accuracy | Macro F1 | Precision | Recall |
|---|---|---|---|---|---|---|
| 4.0× | Real | SVM | 0.797 | 0.797 | 0.813 | 0.797 |
| | Real | NN | 0.802 | 0.802 | 0.817 | 0.802 |
| | Real+Synth. | SVM | 0.806 | 0.803 | 0.823 | 0.807 |
| | Real+Synth. | NN | **0.819** | **0.817** | **0.830** | **0.819** |
| 2.0× | Real | SVM | 0.855 | 0.857 | 0.867 | 0.857 |
| | Real | NN | 0.863 | 0.863 | 0.872 | 0.863 |
| | Real+Synth. | SVM | 0.860 | 0.863 | 0.860 | 0.857 |
| | Real+Synth. | NN | **0.891** | **0.894** | **0.906** | **0.891** |

Table 1: Results on collection 1952. Downsampling reflects the down-scaling of brain volumes. Input represents the input of a classifier: Real means only use real training data, while 'Real+Synth.' means real training data plus synthetic data which is produced by ICW-GAN. Two classifiers are utilized: SVM and neural networks (NN). Results show that augmenting real data with synthetic data improves classification performance.

## 3.2 3D AUXILIARY CLASSIFIER AND DISCRIMINATOR GAN (ACD-GAN)

We also develop a conditional GAN architecture which jointly trains the generator, the discriminator and the classifier. This form of training is inspired by the AC-GAN (Odena et al., 2016) formulation. Our generator architecture is illustrated in Figure 1 and identical to the one employed for the ICW-GAN. The combination of classifier and discriminator is provided in Figure 2. As before, the discriminator assesses the logit of the probability that a provided sample is real or artificially generated. In addition, the classifier provides a probability distribution over the class labels. In this model, training passes labels through the generator and the discriminator in order to condition the adversarial process. However, no labels are employed in the classification stream. The classifier loss, $L_C$, is the log-likelihood of the correct class. Compared to $L$ in Eq. (4), the discriminator loss, $L_D$ includes the additional classifier component

$$L_C = \mathbb{E}_{\boldsymbol{x} \sim P_{data}(\boldsymbol{x})}[\log P(C = c|\boldsymbol{x})] + \mathbb{E}_{\boldsymbol{z} \sim P_{\boldsymbol{z}}(\boldsymbol{z})}[\log P(C = c|G(\boldsymbol{z})]. \tag{5}$$

The generator loss function, $L_G$, consists of the likelihood of fake images and one part of $L_C$, i.e.,

$$L_G = \mathbb{E}_{\boldsymbol{z} \sim P_{\boldsymbol{z}}(\boldsymbol{z})}[P(D = \text{fake}|G(\boldsymbol{z})) + \mathbb{E}_{\boldsymbol{z} \sim P_{\boldsymbol{z}}(\boldsymbol{z})}[P(C = c|G(\boldsymbol{z})]. \tag{6}$$

Similarly, the discriminator loss function is given by $L_D = L + L_C$. We optimize the generator, classifier and discriminator loss functions using Adam optimizer (Kingma & Ba, 2014).

## 4 EXPERIMENTS

We examine the performance of the two aforementioned models on three different datasets from Neurovault (Gorgolewski et al., 2015): collections 1952, 2138 and 503[1]. To evaluate the benefits

---

[1]The collections are publicly available online, e.g., https://neurovault.org/collections/503

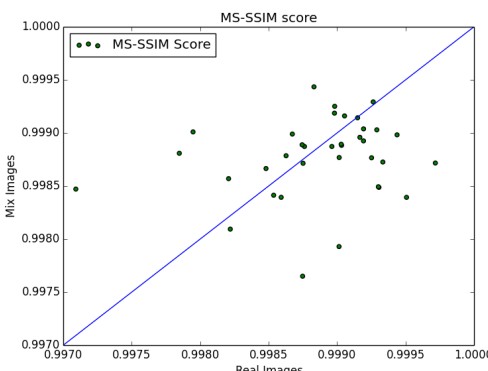 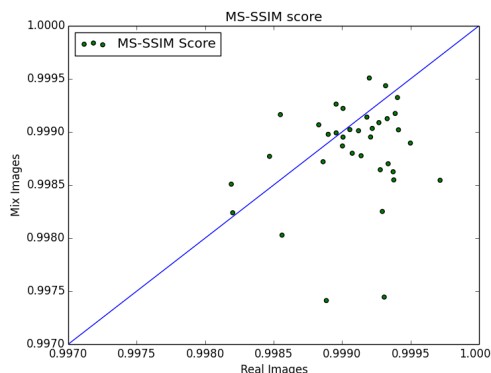

Figure 4: Mean MS-SSIM scores between pairs of images within a given class for brain volumes of 4.0× downsampling (left) and 2.0× downsampling (right). We calculated the MS-SSIM scores on low resolution data of collection 1952 and synthetic data was generated using the ICW-GAN. Each point represents an individual class. Values in horizontal axis are MS-SSIM scores computed on real brain images, while values on the vertical axis are calculated using mixed images.

| Downsampling | Input | Classifier | Accuracy | Macro F1 | Precision | Recall |
|---|---|---|---|---|---|---|
| 8.0× | Real | SVM | 0.523 | 0.480 | 0.497 | 0.523 |
| | Real | NN | 0.530 | 0.517 | 0.545 | 0.530 |
| | Real+Synth. | SVM | 0.531 | 0.493 | 0.510 | 0.533 |
| | Real+Synth. | NN | **0.562** | **0.539** | **0.568** | **0.563** |
| 4.0× | Real | SVM | 0.555 | 0.507 | 0.517 | 0.533 |
| | Real | NN | 0.723 | 0.712 | **0.737** | **0.723** |
| | Real+Synth. | SVM | 0.562 | 0.517 | 0.527 | 0.563 |
| | Real+Synth. | NN | **0.737** | **0.715** | 0.727 | **0.723** |

Table 2: Results on collection 2138. The synthetic images used here were generated from the ICW-GAN. Input represents the input of a classifier: Real means only use real training data, while 'Real+Synth.' means real training data plus generated data. Two classifiers are utilized: SVM and neural networks (NN). Results show that augmenting real data with synthetic data improves classification performance.

of generated fMRI data for classifier training, we focus on two variants of the model architecture for generating images. The original images are downsampled using the nilearn python package[2] to $13 \times 15 \times 11$ (lower resolution) and $26 \times 31 \times 23$ (higher resolution) spatial resolutions for collection 1952, and $13 \times 15 \times 11$ (lower resolution) and $26 \times 31 \times 22$ (higher resolution) resolutions for the other two collections.

In Section 3, we present the architectures of our models for generating low resolution data, which only have 3 convolutional layers. For high resolution data we add one more convolutional and deconvolutional layer in the discriminator and generator respectively. For high resolution data, the classifier also uses 4 convolutional layers in the ACD-GAN. We first show qualitative results for generated 3D volumes, then we provide quantitative results for 3D volume classification. We use cross validation in all reported results to carefully assess the classifier performance. See the supplementary material for details.

**Dataset 1: Collection 1952**  Collection 1952 has 6573 brain images, 45 classes with a total number of 19 sub-classes. The labels include task description that subjects were going through during the imaging process, such as visual, language and calculate. A combination of 2 to 4 sub-classes are subsumed for a class. There are a total of 45 classes because not every potential combination is observed. In our experiments, we work on a subset of 36 classes because 9 of the 45 classes have less than 30 examples. If a class has more than 100 images, we split images in that class 7:1:2 into training, validation and test datasets. Below 100 images but more than 30 images we use a 3:1:2 data split. Note that we ensure that the test data always consists of real images only. We train the ICW-GAN and ACD-GAN for 2500 epochs with a learning rate of 1e-4 and 50% exponential decay for each 3000 iterations and a batch size of 50. 2-D projections of several brain volumes generated by ICW-GAN are illustrated in Figure 3. The projections in the top row of Figure 3 are brains of

---

[2]http://nilearn.github.io

| Downsampling | Input | Classifier | Accuracy | Macro F1 | Precision | Recall |
|---|---|---|---|---|---|---|
| 6.0× | Real | SVM | 0.212 | 0.210 | 0.220 | 0.210 |
| | Real | NN | 0.293 | 0.287 | 0.302 | 0.293 |
| | Real+Synth. | SVM | 0.225 | 0.227 | 0.24 | 0.223 |
| | Real+Synth. | NN | **0.299** | **0.298** | **0.321** | **0.298** |
| 3.0× | Real | SVM | 0.233 | 0.230 | 0.233 | 0.233 |
| | Real | NN | 0.358 | **0.386** | 0.390 | 0.358 |
| | Real+Synth. | SVM | 0.231 | 0.227 | 0.230 | 0.230 |
| | Real+Synth. | NN | **0.373** | 0.380 | **0.410** | **0.372** |

Table 3: Results on collection 503. Similarly, downsampling reflects the down-scaling of brain volumes. Input represents the input of a classifier: Real means only use real training data, while 'Real+Synth.' means real training data plus generated data. Two classifiers are utilized: SVM and neural networks (NN). Results show that augmenting real data with synthetic data improves classification performance.

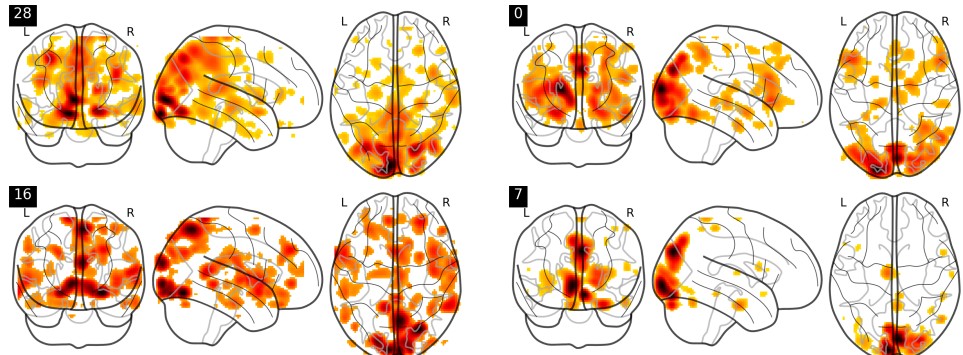

Figure 5: 2-D projections of synthetic brain volumes by ACD-GAN using training data from collection 503 (4.0× downsampling). The left top tag is the class generated brain belongs to (details in the supplement). Each class represents a picture being shown to the subjects. Linear normalization is used and the lower threshold is 0.3.

class 'visual words, language, visual,' and those at the bottom are from class 'places, visual, right hand.' By examining the generated brain images, we find diversity in both intra-class and between-class. We refer the interested reader to the supplementary material for a comparison of generated and real brain images.

Inspired by Odena et al. (2016), we compute the multi-scale structural similarity (MS-SSIM) (Wang et al., 2003) to examine the intra-class diversity of the data generated by ICW-GAN. The MS-SSIM score is a similarity metric based on image similarity at different resolutions. Its values are within $[0, 1]$. Higher MS-SSIM values correspond to more similar images. We measure the mean MS-SSIM score with 100 randomly chosen volumes of mixed data with a given class, and ones of real training data. In $13 \times 15 \times 11$ resolution, we find that 22 classes with mixed data have a lower MS-SSIM score than only with real training data. In other words, 61.1% classes with mixed data have sample variability that exceeds those only with real training data. In $26 \times 31 \times 23$ resolution, 69.4% classes with mixed data have a lower MS-SSIM score. See Figure 4 for details.

To further assess the quality of the generated data we evaluate classification performance. Note that the test data is always composed of real images. The classification results are shown in Table 1. Downsampling refers to the downscaling factor as already discussed. The second column indicates the type of training data we use for the classifier: only using real data ('Real'), or using the mixed data of real and generated volumes ('Real+G'). The third column denotes the classifier type, i.e., an SVM or a deep net ('NN'). We use the validation dataset to choose the best training models and use these models to classify the test data. We observe the deep net classifier to generally outperform SVMs and we observe generated data to be beneficial for classifier training.

**Dataset 2: Collection 2138** For data in collection 2138, there are 1847 brain images, 61 classes and 50 labels. Because of the small size of the dataset, we randomly choose 70% of the brain images as training data and leave 30% as test data. In this case, we do not have development data to supervise the training process; thus, we train our models for 1000 epochs in several runs and record the best classification results, which are summarized in Table 2. In this collection, we downsample brains by a factor of 8 and 4 to volume sizes of $13 \times 15 \times 11$ and $26 \times 31 \times 22$ respectively. Similar to the

| Dataset | Model | Accuracy | Macro F1 | Precision | Recall |
|---|---|---|---|---|---|
| 1952 | AC-GAN | 0.800 | 0.741 | 0.784 | 0.745 |
| | ICW-GAN | 0.819 | **0.817** | **0.830** | **0.819** |
| | ACD-GAN | **0.835** | 0.815 | 0.828 | 0.817 |
| 2138 | ICW-GAN | 0.562 | 0.539 | 0.568 | **0.563** |
| | ACD-GAN | **0.569** | **0.547** | **0.579** | 0.560 |
| 503 | ICW-GAN | 0.299 | 0.298 | 0.321 | 0.298 |
| | ACD-GAN | **0.326** | **0.299** | **0.329** | **0.301** |

Table 4: Results of ACD-GAN on the three datasets in the low resolution setting. We find the ACD-GAN to perform slightly better than the ICW-GAN.

earlier reported results we observe deep nets to outperform SVMs, but more importantly, generated data during training was again suitable to improve the classifier performance on real data.

**Dataset 3: Collection 503**   5067 brain images are in collection 503. Subjects in this dataset are required to respond to 30 images from the International Affective Picture Set (Chang et al., 2015). These 30 images were used to train the Picture Induced Negative Emotion Signature also described in this paper. We consider the 30 pictures classes of brain data. In other words, there are overall 30 classes and each class represents a picture shown to subjects. This collection needs extra pre-processing because brain volumes are available with two shapes: $79 \times 95 \times 68$ and $91 \times 109 \times 91$. We conform large volumes to the small ones. Similarly, all experiments are at the two levels of resolution. Classification results are summarized in Table 3 and again follow the trend reported earlier.

**Results of Using ACD-GAN**   Since our ICW-GAN achieves the best performance in the three databases when we use mixed training data, we compare these best results to results of ACD-GAN at the low resolution. 2-D projections of brain volumes generated by ACD-GAN are shown in Figure 5. In Table 4, we present classification results of our models and show that ACD-GAN obtains improved accuracy scores compared to a ICW-GAN for most of the investigated metrics. As a baseline, we also implemented the recent generative model – the 3-D AC-GAN Odena et al. (2016). According to all the results presented above, the proposed methods achieve much improved overall better performance on the three datasets. This is true at both levels of resolution for brain images. The results also outperform a 3-D AC-GAN applied to the same dataset.

We also observe that our strategy of mixing real and generated data does not always yield the best results. For example, using real data only to train the NN classifier achieves the largest macro F1 score at high resolution in Table 3. However, the proposed approach consistently improves accuracy metrics – which is by far the most popular metric for evaluating multi-class classification performance.

## 5   CONCLUSIONS

Generative models provide a useful tool for understanding the individual variability of brain images. The results of this manuscript show – to our knowledge for the first time, that 3-D conditional GANs, in particular our proposed ICW-GAN and ACD-GAN, can generate high quality diverse and task dependent brain images. We hope our results inspire additional research on generative models for brain imaging data. Beyond qualitative evaluation, we evaluate quantitative performance by using the generated images as additional training data in a predictive model – mixing synthetic and real data to train classifiers. The results show that our synthetic data augmentation can significantly improve classification accuracy – a result which may be of independent interest. Future work will focus on additional qualitative evaluation of the generated images by neuroscience experts and exploration of various applications. We also plan to more throughly investigate the trained models to further explore what it may contribute to the science of individual variability in neuroimaging. Finally, we plan to expand our models to combine data across multiple studies – each of which use different labels, by exploring techniques for merging labels based on the underlying cognitive processes (Poldrack, 2006).

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

## 6 SUPPLEMENTARY MATERIAL

### 6.1 CROSS-VALIDATION STRATEGY

First, we explain our cross-validation strategy. In Figure 6, the blue bar represents training data and the green bar is test data. For 3-fold cross-validation, first we partition test data to three folds and each fold maintains a similar proportion of data for each class as original test data. For each round of cross-validation, we leverage one of the three parts as test data and left two folds together with training data to train a model. The same strategy is used with generated data to ensure that test performance is only based on the test data. Each evaluation metric is the average of the three rounds.

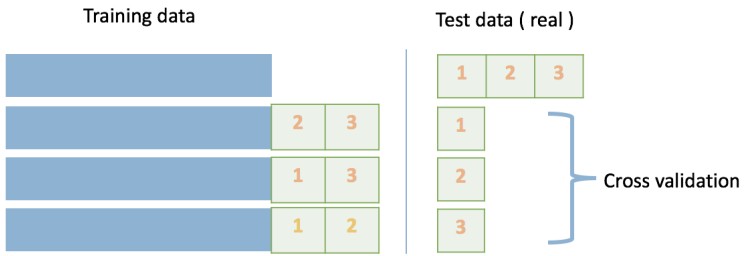

Figure 6: Cartoon illustration of our 3-fold cross validation strategy.

### 6.2 VARIANCE OF CROSS-VALIDATED PERFORMANCE

We tested our model with various cross-validation settings and calculated the variances of the evaluation metrics (Table 5). We conducted this experiment on low resolution data from collection 1952 and synthetic data generated by the ICW-GAN. Except for the number of folds used when partitioning the test dataset, the training strategy for 5-fold and 10-fold cross-validation is similar to that of 3-fold cross-validation. The small variances suggest that the reported accuracy differences, while small, are indeed significant.

| N-fold | Accuracy | Var_Acc | F1 | Var_F1 | Precision | Var_P | Recall | Var_R |
|--------|----------|---------|-----|--------|-----------|-------|--------|-------|
| 3-fold | 0.819 | 0.0002782 | 0.817 | 0.0002543 | 0.830 | 0.0001843 | 0.819 | 0.0002432 |
| 5-fold | 0.8368 | 0.0002432 | 0.841 | 0.0001985 | 0.862 | 0.0001643 | 0.837 | 0.0002434 |
| 10-fold | 0.843 | 0.0002782 | 0.857 | 0.0003027 | 0.894 | 0.0004785 | 0.843 | 0.0003789 |

Table 5: Accuracy, F1, Precision, Recall and their variance (column 3,5,7,9) for 3-fold, 5-fold and 10-fold cross validation. We conducted this experiment with the training data of mixed 'Real+Synth.' data of collection 1952 in the low resolution ($4.0\times$) setup.

### 6.3 TRAINING LOSS CURVE OF ICW-GAN IN VALIDATION DATASET IN COLLECTION 1952

We present the ICW-GAN training loss curve evaluated on the validation dataset of collection 1952 in Figure 7. This illustrates the stability of the proposed improved Wasserstein variants.

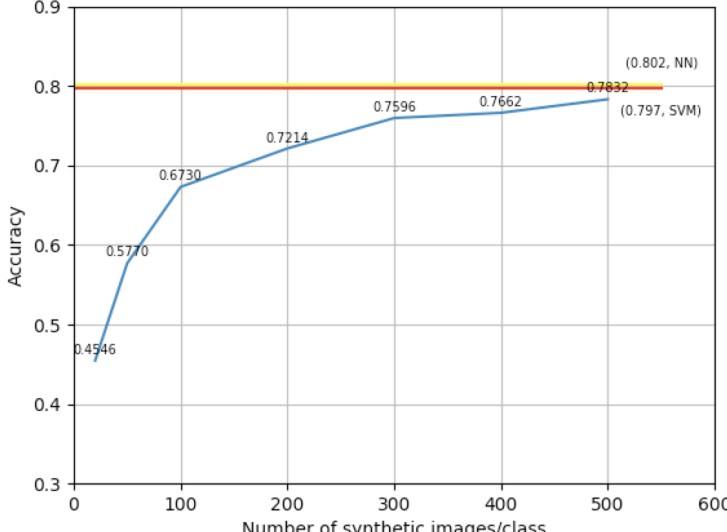

Figure 8: This figure shows the trajectory of multiclass accuracy along with the number of artificial training samples resembling collection 1952 at the low resolution setup. In this case, we only use generated data from the ICW-GAN to train the deep net classifier. The horizontal axis represents the number of artificial data used for each class, while the vertical axis represents the accuracy value. Two light brown lines note the accuracy obtained by only using real data with the SVM and NN classifier.

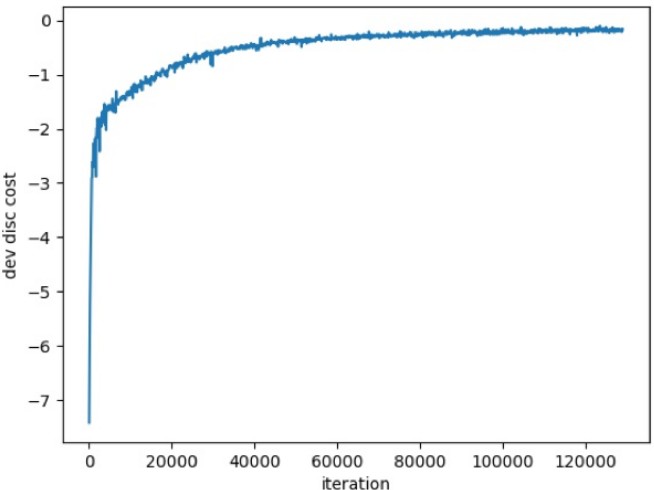

Figure 7: Training loss curve of the ICW-GAN on the low-resolution validation dataset of collection 1952.

### 6.4 RESULTS OF ONLY USING GENERATED DATA FOR TRAINING

In order to further evaluate the quality of the generated data for collection 1952, we trained using only generated data and tested with real data. We used the deep net classifier for evaluation in this case while varying the number of input samples for each class. Figure 8 shows that in general, the test accuracy improves as the amount of artificial training data increases. The red and yellow lines show the accuracy obtained when only using real data. These results suggest that the generated data can be as effective for classification as real data with sufficient numbers of samples.

| Training data | Accuracy | F1 | Precision | Recall |
|---|---|---|---|---|
| Synth. data from GMM (20 images/class) | 0.203 | 0.309 | 0.309 | 0.202 |
| Synth. data from GMM (500 images/class) | 0.720 | 0.725 | 0.765 | 0.720 |
| Real+Synth. (from GMM) | 0.793 | 0.798 | 0.824 | 0.793 |
| Synth. data from ICW-GAN (20 images/class) | **0.458** | **0.433** | **0.537** | **0.458** |
| Synth. data from ICW-GAN (500 images/class) | **0.783** | **0.776** | **0.805** | **0.783** |
| Real+Synth. (from GAN) | **0.819** | **0.817** | **0.830** | **0.819** |

Table 6: Results of synthesizing data using collection 1952 (low resolution). Comparison between the GMM and ICW-GAN. We list 6 training data strategies: in the 1st, 2nd , 4th and 5th row, we only use synthetic data to train the deep net classifier while in the 3rd and 6th row, we mix real and synthetic data together to train the same classifier.

We also used a Gaussian Mixture Model (GMM), i.e., trained separately for each class, as a simple baseline generative model. The results are shown in Table 6, where 6 training strategies were listed and the deep net classifier (same as the NN classifier) was used for all the strategies. I.e., we leveraged 20 synthetic images in each class ($20 \times$ number of classes in all) to train the NN classifier. Note that both generative models learned the probability distribution from collection 1952. The evaluation scores of our models are significantly higher than the ones obtained with a GMM, particularly when using only synthetic data to train the classifier.

### 6.5 GENERATED IMAGE QUALITATIVE ANALYSIS

We used NeuroSynth[3], a brain decoding tool to qualitatively analyze generated volumes. Its principle is to compare activity regions of a given brain volume to a data pool and then to obtain an analysis report about correlation to specific activity patterns. Figure 9 shows a generated brain image of collection 1952 by the ICW-GAN in low resolution and its analysis report with the top 10 ranking list of correlation. The supervised class of this generated brain is "Non-human sound, auditory", and several correlations with top rank are related to auditory and sound.

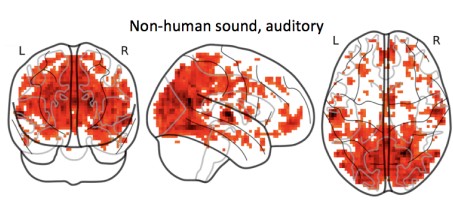

| Analysis | Correlation |
|---|---|
| auditory cortex | 0.207 |
| auditory | 0.198 |
| heschi | 0.195 |
| heschl gyrus | 0.194 |
| sounds | 0.190 |
| fractional anisotropy | 0.189 |
| fa | 0.188 |
| pitch | 0.187 |
| anisotropy fa | 0.186 |
| anisotropy | 0.185 |

Figure 9: The projections of a generated brain volume (left) and its corresponding analysis report (right). The volume belongs to the class 'Non-human sound, auditory.'

### 6.6 ADDITIONAL MULTICLASS CLASSIFICATION RESULTS

We also examine our models using a multilabel classification strategy in low resolution data of collection 1952. As described before, one class in collection 1952 consists of several labels and the collection has a total of 19 sub-labels. Instead of encoding the classes with a one-hot vector of 45 dimensions, we encode them in 19 dimensions, each of which represents a sub-label. The value in a dimension is 1 only if the sample possesses that label, otherwise, 0, i.e., a sample comes with a 19 dimensional binary vector. This representation is demanding since the probability space is significantly larger, specifically, $2^{19}$ possibilities, for a classifier to learn and discriminate. We use SVM to train real and mixed training data (synthetic data obtained from ICW-GAN), both of which

---

[3]The webpage of NeuroSynth is http://neurosynth.org/decode/

| Input | Accuracy | Macro F1 | Precision | Recall |
|---|---|---|---|---|
| Real | 0.4838 | 0.57 | 0.76 | 0.48 |
| Real+Synth. | **0.4992** | 0.57 | 0.75 | 0.49 |

Table 7: Multilabel results for collection 1952 at 4.0× downsampling and synthetic data were from the ICW-GAN. Even in this highly demanding task, accuracy scores with mixed data (the second row) outperform the baseline (the first row).

are in the resolution of $13 \times 15 \times 11$, and utilize 3-fold cross-validation as discussed in Section 6.1. Several evaluation scores are shown in Table 7. We observe that even in this highly demanding task, accuracy scores with mixed data (the second row) outperform the baseline (the first row).

## 6.7 REAL BRAIN IMAGES OF COLLECTION 1952 WITH LABELS ON TOP

We present several real images of collection 1952 with their labels above the 2D projections of brain images in Figure 10.

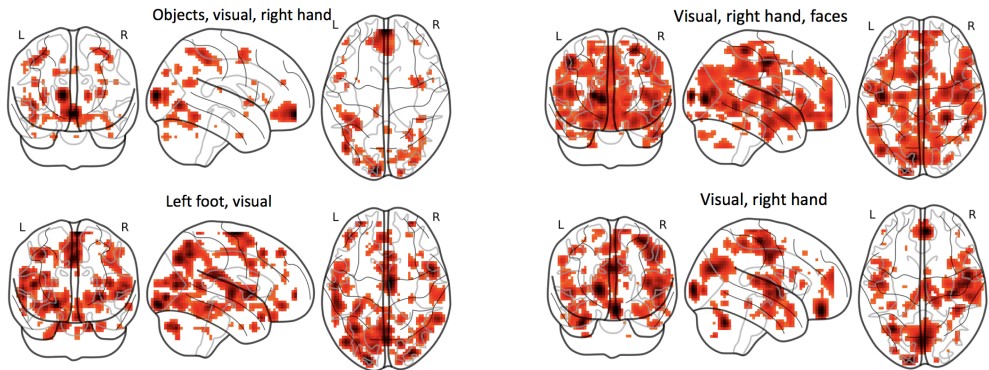

Figure 10: Real images in collection 1952 at 4.0× downsampling with labels on top. Linear normalization is used and the lower threshold for plotting these images is 0.48.

## 6.8 ADDITIONAL SYNTHETIC BRAIN IMAGES

More synthetic brain images using ICW-GAN are shown in this section. Figures 11-13 show generated ones of low resolution from collection 1952, 2138 and 503 respectively. For collection 1952, classes are given above the images.

As for collection 2138, the corresponding categories of the classes are as follows:

- **14:** visual form recognition, feature comparison, response selection, response execution, relational comparison, visual pattern recognition
- **22:** response execution, working memory, body maintenance, visual body recognition
- **4:** response selection, response execution, punishment processing
- **43:** motion detection
- **2:** response selection, response execution, animacy perception, animacy decision, motion detection
- **3:** response selection, response execution, motion detection

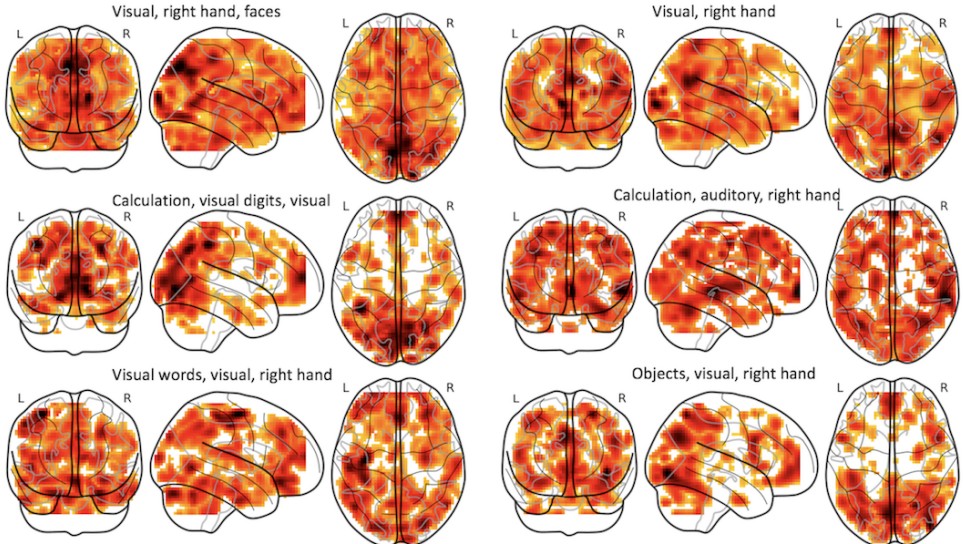

Figure 11: $4.0\times$ downsampled brain images in Collection 1952 generated by ICW-GAN with labels on top. The lower threshold for plotting these images is 0.3.

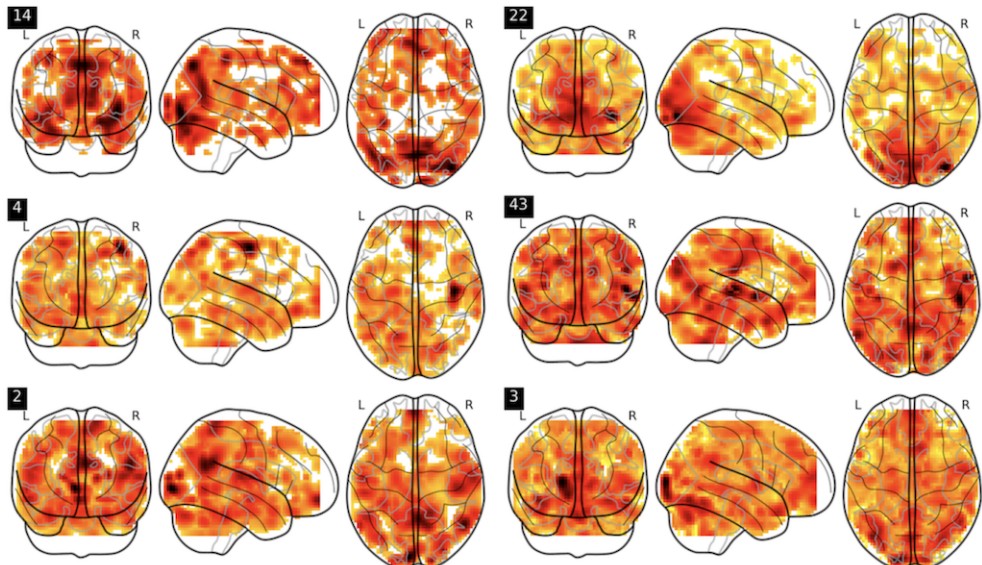

Figure 12: $8.0\times$ downsampled brain images in collection 2138 generated by ICW-GAN. A left top tag is the class generated brains belong to and we list their corresponding categories in the supplementary material as well.

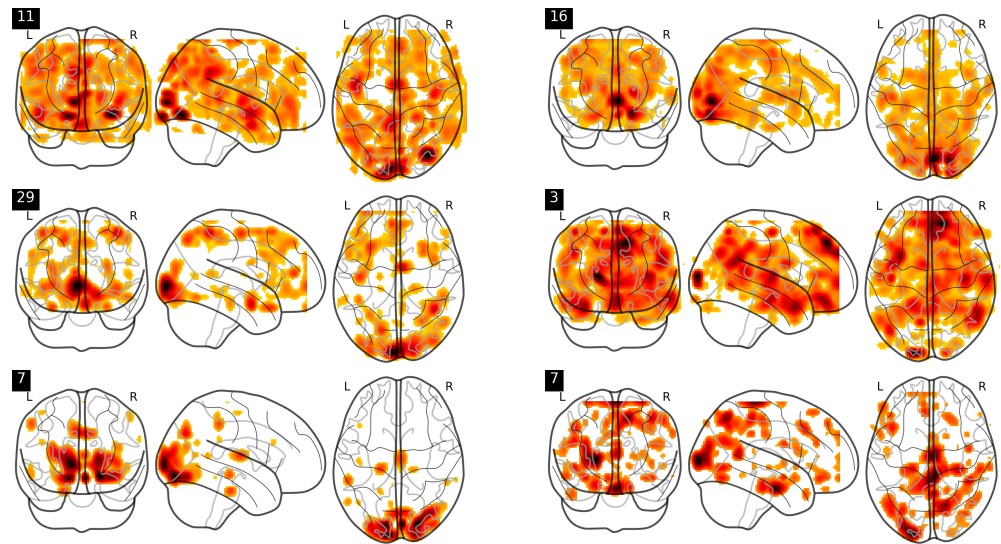

Figure 13: 6.0× downsampled brain images in collection 503 generated by ICW-GAN and a left top tag represents the stimulating picture for subjects.

