# OpenReview forum: "Hallucinating brains with artificial brains"
_ICLR.cc/2018/Conference — Reject_

### Official Review · AnonReviewer3 · 2017-11-24
**GANS on neuroimaging data**

**Rating:** 8
**Confidence:** 5

**Review:**

Quality

This is a very clear contribution which elegantly demonstrates the use of extensions of GAN variants in the context of neuroimaging.

Clarity

The paper is well-written. Methods and results are clearly described. The authors state significant improvements in classification using generated data. These claims should be substantiated with significance tests comparing classification on standard versus augmented datasets.

Originality

This is one of the first uses of GANs in the context of neuroimaging.

Significance

The approach outlined in this paper may spawn a new research direction.

Pros

Well-written and original contribution demonstrating the use of GANs in the context of neuroimaging.

Cons

The focus on neuroimaging might be less relevant to the broader AI community.

---

> ### Author Response · Authors · 2018-01-05
> **Rebuttal**
>
> Thank you for your strong review.
>
> - w.r.t. relevance to the AI community: We think neuroscience is an integral part of the larger AI community, benefiting both sides when seeking inspiration. Further, we expect that many of the techniques we propose are directly applicable to more common computer vision tasks.

---

### Official Review · AnonReviewer1 · 2017-11-26
**Interesting idea with some potential contribution on GAN architectures to generate fMRI scans**

**Rating:** 6
**Confidence:** 4

**Review:**

This paper proposes to use 3D conditional GAN models to generate
fMRI scans. Using the generated images, paper reports improvement
in classification accuracy on various tasks.

One claim of the paper is that a generative model of fMRI
data can help to caracterize and understand variability of scans
across subjects.

Article is based on recent works such as Wasserstein GANs and AC-GANs
by (Odena et al., 2016).

Despite the rich literature of this recent topic the related work
section is rather convincing.

Model presented extends IW-GAN by using 3D convolution and also
by supervising the generator using sample labels.

Major:

- The size of the generated images is up to 26x31x22 which is limited
(about half the size of the actual resolution of fMRI data). As a
consequence results on decoding learning task using low resolution
images can end up worse than with the actual data (as pointed out).
What it means is that the actual impact of the work is probably limited.

- Generating high resolution images with GANs even on faces for which
there is almost infinite data is still a challenge. Here a few thousand
data points are used. So it raises too concerns: First is it enough?
Using so-called learning curves is a good way to answer this. Second
is what are the contributions to the state-of-the-art of the 2
methods introduced? Said differently, as there
is no classification results using images produced by an another
GAN architecture it is hard to say that the extra complexity
proposed here (which is a bit contribution of the work) is actually
necessary.

Minor:

- Fonts in figure 4 are too small.

---

> ### Author Response · Authors · 2018-01-05
> **Rebuttal**
>
> Thank you for the clear review.
>
> - w.r.t. size of generated brain maps: Decreasing the resolution of the imaging data is common practice in the neuroimaging analysis, e.g., it is built into the Nilearn python package. Interestingly and in contrast to the reviewer comment, we observe benefits by including synthetic data with higher resolution.
>
> - w.r.t. effectiveness of generative model(s): To highlight the effectiveness of the proposed models, we have added additional results for two generative models to the revised manuscript, the AC-GAN (Tab. 4) and Gaussian Mixture Model (Tab. 6). Our results show that both AC-GAN and GMM achieve much worse results. To further evaluate the generative model, we experimented with using only generated data to train the classifiers (Fig. 8). Our results in the revised manuscript suggest that using several hundred artificial images per class is comparable to using real images.
>
> - w.r.t. stability of GAN: To demonstrate the stability we added training loss curves to the revised manuscript (Fig. 7). We did not observe any issues which we attribute to the stability of Wasserstein variants.

---

### Official Review · AnonReviewer2 · 2017-11-29
**Relevant hard problem and poor evidence for proposed solution**

**Rating:** 5
**Confidence:** 3

**Review:**

The work is motivated by a real challenge of neuroimaging analysis: how to increase the amount of data to support the learning of brain decoding.
The contribution seems to mix two objectives: on one hand to prove that it is possible to do data augmentation for fMRI brain decoding, on the other hand to design (or better to extend) a new model (to be more precise two models).
Concerning the first objective the empirical results do not provide meaningful support that the generative model is really effective. The improvement is really tiny and a statistical test (not included in the analysis) probably wouldn't pass a significant threshold.  This analysis is missing a straw man. It is not clear whether the difference in the evaluation measures is related to the greater number of examples or by the specific generative model.
Concerning the contribution of the model, one novelty is the conditional formulation of the discriminator. The design of the empirical evaluation doesn't address the analysis of the impact of this new formulation. It is not clear whether the supposed improvement is related to the conditional formulation.
Figure 3 and Figure 5 illustrate the brain maps generated for Collection 1952 with ICW-GAN and for collection 503 with ACD-GAN. It is not clear how the authors operated the choices of these figures. From the perspective of neuroscience a reader,  would expect to look at the brain maps for the same collection with different methods. The pairwise brain maps would support the interpretation of the generated data. It is worthwhile to remember that the location of brain activations is crucial to detect whether the brain decoding (classification) relies on artifacts or confounds.

Minor comments
- typos: "a first application or this" => "a first application of this" (p.2)
- "qualitative quality" (p.2)

---

> ### Author Response · Authors · 2018-01-05
> **Rebuttal**
>
> Thank you for your valuable comments on our paper.
>
> - w.r.t. goals of the manuscript: From a neuroscience perspective, the paper develops a mechanism addressing two concerns: (i) how to generate synthetic samples which help address the shortage of data that is common in neuroimaging, and can be used to analyze inter-individual variability, among other applications; (ii) how to evaluate artificially generated neuroimaging data.
>
> - w.r.t. effectiveness of generative models: To highlight the effectiveness of the proposed models, we have added additional results for two generative models to the revised manuscript, the AC-GAN (Tab. 4) and Gaussian Mixture Model (Tab. 6). Our results show that both AC-GAN and GMM achieve much worse results. To further evaluate the generative model, we experimented with using only generated data to train the classifiers (Fig.8). Our results in the revised manuscript suggest that using several hundred fake images per class is comparable to using real images.
>
> - w.r.t. improvements to classification performance: We point the reviewer to Tab. 4 for a comparison of different GAN architectures. Also, we mention that for the results reported in Tab. 1-3, both SVM and deep net classifiers are compared with and without artificially generated data. We think this clearly demonstrates the benefits of adding data obtained from GANs. The high performance of the baselines, resulting from careful tuning, are easily on par with typically reported numbers in the literature. Moreover, we point out that the reported improvements are consistent across a variety of metrics. This experimental evaluation suggests that the reported improvements aren’t small and are hard to achieve. In Tab. 5 of the revised manuscript, we provide the variance of the cross-validated performance. The small variances suggest the significance of the performance differences.
>
> - w.r.t. illustrated brain maps and brain decoding: We refer the reviewer to Fig. 5 in the manuscript and Fig. 11-13 in the supplementary material for additional results. We have clarified details as requested, for instance, Fig. 5 and Fig. 13 show the synthetic images with label ‘7’ in collection 503 by the ACD-GAN and ICW-GAN respectively.

---

### Author Response · Authors · 2018-01-05
**To all reviewers: Thank you for the valuable comments!**

We thank all reviewers for their constructive feedback and address their comments in the following. We  will release all code soon.

---

### Decision · Program_Chairs · 2018-01-29
**ICLR 2018 Conference Acceptance Decision**

**Decision:**

Reject

**Comment:**

The submission proposes to use GANs to learn a generative model of fMRI scans that can then be used for downstream classification tasks.  Although there was some appreciation from the reviewers of the approach, there were several important remaining concerns:

1) From Reviewer 1: "Generating high resolution images with GANs even on faces for which there is almost infinite data is still a challenge. Here a few thousand data points are used. So it raises too concerns: First is it enough?"

and

2) R1 and R2 both raised concerns about the significance of the improvements.  Looking through the tables, there are many reported differences that are reasonably small, and no error bars or significance are given.  This should be a requirement for an empirical paper about fMRI.